# Risk Factors of Infection in Relapsed/Refractory Multiple Myeloma Patients Treated with Lenalidomide and Dexamethasone (Rd) Regimen: Real-Life Results of a Large Single-Center Study

**DOI:** 10.3390/jcm11195908

**Published:** 2022-10-07

**Authors:** Damian Mikulski, Paweł Robak, Wiktoria Ryżewska, Kamila Stańczak, Kacper Kościelny, Joanna Góra-Tybor, Tadeusz Robak

**Affiliations:** 1Department of Hematooncology, Copernicus Memorial Hospital, Comprehensive Cancer Center and Traumatology, 93-510 Lodz, Poland; 2Department of Biostatistics and Translational Medicine, Medical University of Lodz, 90-419 Lodz, Poland; 3Department of Experimental Hematology, Medical University of Lodz, 90-419 Lodz, Poland; 4Department of Hematology, Medical University of Lodz, 90-419 Lodz, Poland; 5Department of General Hematology, Copernicus Memorial Hospital, 93-510 Lodz, Poland

**Keywords:** complication, dexamethasone, IMiD, infection, lenalidomide, multiple myeloma, neutropenia, Rd

## Abstract

Lenalidomide-based regimens are effective treatment options for patients with relapsed/refractory multiple myeloma (RRMM). However, they are associated with an increased risk of infectious complications. This study examines the clinical factors influencing the occurrence of infection in MM patients treated with lenalidomide and dexamethasone (Rd). A retrospective analysis of all patients who received the Rd regimen between 2017 and 2021 at our institution was performed. The study group consisted of 174 patients and the median age was 65 years. Most patients (n = 110, 63.2%) received the Rd treatment in second-line treatment. The majority of patients (64.3%) received bortezomib-based regimens in the first line of treatment. The median progression-free survival was 12.6 (95% CI: 9.5–16.2) months, and the median overall survival was 22.3 (95% CI: 15.9–28.6) months. The overall response rate was 64.1%, 12.7% of patients achieved complete response, and 20.4% had a very good partial response. In multivariate logistic regression analysis, hypoalbuminemia (OR 4.2, 95% CI: 1.6–11.2, *p* = 0.0039), autologous hematopoietic stem cell transplantation (AHSCT) before Rd (OR 2.6, 95% CI: 1.0–6.7, *p* = 0.048), and anemia grade ≥3 (OR 5.0, 95% CI: 1.8–14.0, *p* = 0.002) were independent factors related to the occurrence of infections. In conclusion, in this large cohort of RRMM patients, AHSCT before Rd regimen therapy, hypoalbuminemia, and anemia during treatment were identified as three independent factors influencing the frequency of infections during Rd therapy. Patients with established risk factors may benefit from optimal supportive therapy.

## 1. Introduction

Multiple myeloma (MM) is a hematological malignancy characterized by the abnormal growth of monoclonal plasma cells, renal impairment, hypercalcemia, bone lesions, and anemia [1,2]. It is the second most common type of blood cancer in the United States and Europe; frequently develops in people over the age of 60, with the median age at diagnosis being 69 in the US; and is more prevalent in men than women [3,4]. Although MM remains incurable, patient survival rates have considerably increased over the last two decades due to the introduction of novel drugs such as proteasome inhibitors (PI), immunomodulating agents (IMiD), and monoclonal antibodies [5].

Lenalidomide is an IMiD and a molecular analog of thalidomide. It was registered in 2006 by the U.S. Food and Drug Administration (FDA) in MM patients previously treated with at least one line of therapy [6,7]. Lenalidomide directly binds to cereblon (CRBN), the component of CRL4CRBN E3 ubiquitin ligase, inducing the ubiquitination and degradation of IKZF1 (Ikaros) and IKZF3 (Aiolos) in MM cells, leading to cytotoxicity and immunomodulatory effects [8,9]. IKZF1 and IKZF3 are preferentially ubiquitinated and degraded in the presence of lenalidomide in MM cells, as demonstrated by proteome-wide analyses [10]. Currently, lenalidomide is the backbone of multiple three- and four-drug regimens and is recommended as part of the induction regimen for newly diagnosed MM patients, both eligible and noneligible for autologous stem cell transplantation (ASCT) [11].

MM patients are at a seven times greater risk of bacterial infection, and a 10 times greater risk of viral infection [12]. In the first year following a diagnosis of MM, the risk of some infections, such as pneumonia and septicemia, was nearly ten times that of controls. MM patients have also a considerably increased risk of infection-related mortality compared to age-matched controls [12]. The risk of infection of grade III or higher, pneumonia, and neutropenia exists during all phases of MM therapy. Severe infections are often in the frontline and relapsed/refractory settings [13]. The risk factors for infectious complications in patients with MM can be categorized as patient-related, disease-related, and treatment-related [14]. Patient-related factors include old age, poor performance status, and comorbidities, including diabetes mellitus and renal dysfunction. The disease-related factors include advanced disease (stage III according to the International Staging System), immune dysfunction that includes suppression of cellular and humoral immunity including hypogammaglobulinemia, low CD4+ cell count, and impaired function of natural killer cells [14]. A cumulative number of treatment regimens and corticosteroids are linked with an increased risk of infection in RR MM; the addition of an IMiD such as lenalidomide to corticosteroids has been associated with double the risk of infection compared to steroids alone [15].

The most common adverse effects of lenalidomide include neutropenia, thrombocytopenia, anemia, venous thromboembolism, and infections [16,17]. The latter is commonly tied with neutropenia and long-term use of steroids. Accordingly, infection rates are drastically lower at low dexamethasone doses compared to high doses [18]. However, a metanalysis showed that IMiD-based treatment for relapsed and refractory (RR) MM was associated with a substantially increased risk of severe infection, febrile neutropenia, and pneumonia compared to high-dose dexamethasone [15]. In the pivotal MM-009 and MM-010 trials, the incidence of grade ≥3 pneumonia was 9.1% [19]. Although an increasing number of patients with MM are receiving Rd, little is known about the risk factors for infection, which remain a significant clinical concern in this population.

The aim of this study was to establish clinical factors influencing the occurrence of infection in Rd-treated MM patients. It also reports the real-life Rd treatment outcome results and adverse-effect profile from a large single-institution cohort.

## 2. Patients and Methods

### 2.1. Patients, Treatment, and Adverse Events

This retrospective study included all patients who received the Rd regimen according to the Ministry of Health’s drug reimbursement program for multiple myeloma patients (B.54) between January 2017 and December 2021 at the Department of Hematology, Medical University of Lodz, Poland. Inclusion and exclusion criteria, drug dosing, and monitoring were set according to the reimbursement program. Briefly, the included patients met at least one of the following conditions: treatment with at least two previous lines of therapy, treatment with at least one previous regimen which induced polyneuropathy, or transplant-ineligible patients treated in the first line with the bortezomib-based regimen. The recommended treatment cycle was lenalidomide 25 mg once daily on days 1–21 and dexamethasone 40 mg orally on days 1–4, 9–12, and 17–20 for the first four cycles. After the fourth cycle, 40 mg of dexamethasone was administered only on days 1 to 4. Response to treatment and relapse/progression events were classified according to the International Myeloma Working Group (IMWG) [20]. Adverse events were assessed according to the CTCAE (NCI Common Terminology Criteria for Adverse Events) Version 5.0. In general, infection was detected in patients needing oral or intravenous therapy (antibiotic, antifungal, or antiviral intervention), hospitalization, and/or interruption of Rd medication.

### 2.2. Statistical Analysis

The Shapiro–Wilk test was performed to confirm that the continuous variables had a normal distribution. Normally distributed variables were reported as mean with standard deviation (SD) and non-normally distributed as median with interquartile range (IQR). Survival analysis was performed using a Kaplan–Meier estimate, univariate and multivariate Cox’s proportional hazards models, and the log-rank test. Logistic regression univariate and multivariate analyses were conducted to identify factors influencing the occurrence of infection during Rd treatment. Statistica 13.1 was used for the analyses (TIBCO, Palo Alto, CA, USA). *p*-values below 0.05 were regarded as statistically significant.

## 3. Results

### 3.1. Study Group Characteristics

The study group consisted of 174 patients and the median age was 65 years (IQR, interquartile range: 59.0–71.0). Detailed characteristics of the study cohort are presented in Table 1. Cytogenetics data were available for 52 (29.9%) patients; of these, 30 were identified as high-risk, and 22 as standard-risk according to the IMWG [20,21,22]. The majority of patients (n = 110, 63.2%) received the Rd treatment at the time of the first relapse/progression (second line treatment), 57 patients (32.8%) were treated with the Rd regimen in the third line of treatment, and seven patients (4.0%) received Rd in the fourth line of treatment. The majority of patients (64.3%) received bortezomib-based regimens in the first line of treatment, with the most common being VCD (bortezomib, cyclophosphamide, and dexamethasone) (52.9%). Almost one-third (32.8%) of patients underwent autologous hematopoietic stem cell transplantation (AHSCT) before Rd treatment, and 16 received a transplant after the Rd regimen. The median treatment duration was 8.8 months (IQR: 4.3–18.2). The most common causes of early treatment termination were referral to AHSCT (16 cases), withdrawal of consent to treatment (10 cases), and hematological toxicity (8 cases). Most patients received anti-infectious prophylaxis, including acyclovir (84.1%), 42.0% received sulfamethoxazole/trimethoprim, and 18.2% received fluoroquinolones including ciprofloxacin or levofloxacin. All patients who underwent AHSCT received vaccines according to the local recommendations [23] and international guidelines [24,25], including influenza, hepatitis B, pneumococci, and Haemophilus influenzae.

### 3.2. The Treatment Outcome

The median progression-free survival (PFS) was 12.6 (95% CI: 9.5–16.2) months, and the median overall survival (OS) was 22.3 (95% CI: 15.9–28.6) months. The overall response rate (ORR) was 64.1%, 12.7% of patients achieved complete response (CR), and 20.4% had a very good partial response (VGPR). Univariate Cox regression analysis for PFS and OS regarding potential prognostic factors is shown in Table 2.

In the multivariate Cox regression analysis for PFS, ISS 3 (HR 1.8 95% CI: 1.2–2.7, *p* = 0.0068), high-risk cytogenetics according to IMWG (HR 3.0, 95% CI: 1.7–5.0, *p* = 0.0001), and creatinine >2 mg/dL at commencement of Rd therapy (HR 2.1, 95% CI: 1.1–4.5, *p* = 0.0354) negatively impacted PFS, whereas achievement of CR/VGPR after cycle 6 (HR 0.3 95% CI: 0.1–0.7, *p* = 0.0052) and AHSCT after Rd (HR 0.4, 95% CI: 0.2–0.8, *p* = 0.0104) were independent protective prognostic factors (Figure 1A).

Similarly, International Prognostic System (IPS) 3 (HR 2.1 95% CI: 1.4–3.3, *p* = 0.0012) was related to poorer OS, whereas earlier administration of the Rd regimen (in the second line of treatment) (HR 0.5, 95% CI: 0.2–0.9, *p* = 0.0223) and AHSCT after Rd treatment (HR 0.4, 95% CI: 0.1–1.0, *p* = 0.0398) was related to improved OS (Figure 1B). The corresponding Kaplan–Meier plots are shown in Appendix A.

### 3.3. Adverse Events

The most prevalent nonhematological adverse events in the study group were infections (31.0%), thromboembolic events (11.5%), and polyneuropathy (8.6%). Among infections, the most common were upper respiratory tract infections (14 of 54, 25.9%), followed by pneumonia (12, 22.2%) and urinary tract infections (3, 5.6%). Two patients developed gastrointestinal infections. One patient experienced Varicella zoster virus reactivation, and one patient had Herpes simplex virus infection. One-third of our cohort (58 patients) commenced Rd treatment during the COVID-19 pandemic. Overall, eight patients had confirmed COVID-19 infection, and two required hospitalization. No COVID-19-related death was reported in our study. In 16 cases, the origin of infection could not be established; however, specific treatment, interruption of Rd regimen, and/or hospitalization were needed. The median number of cycles before the infection occurred was two (IQR 1–5 cycles). Overall, we noted that 24 of the 54 observed infections required hospitalization.

In the univariate logistic regression analysis, hypoalbuminemia at Rd initiation, anemia, and neutropenia during Rd treatment were factors increasing the risk of infection (Table 3). Age, ISS, ASCT before Rd, neutropenia and anemia during Rd treatment, and hypoalbuminemia were included in the multivariate analysis. Hypoalbuminemia (OR 4.2, 95% CI: 1.6–11.2, *p* = 0.0039), AHSCT before Rd (OR 2.6, 95% CI: 1.0–6.7, *p* = 0.048), and anemia grade III and IV (OR 5.0, 95% CI: 1.8–14.0, *p* = 0.002) were independent factors related to the occurrence of infections (Figure 2). Among hematological adverse events, the most common were neutropenia grade ≥3 (28.7%), followed by anemia grade ≥3 (23.6%) and thrombocytopenia grade ≥3 (18.4%).

## 4. Discussion

This study examined a large single-center cohort of MM patients treated with the Rd regimen. The findings identify three independent factors that influence the occurrence of infections during treatment: AHSCT before Rd regimen treatment, hypoalbuminemia, and anemia grade ≥3 during treatment. Among the infections, the most common were upper respiratory tract infections, pneumonia, and urinary tract infections. AHSCT is a well-established factor associated with an increased risk of infections [26]. In AHSCT recipients, antibodies against several diseases, including pneumococci, Haemophilus, and measles, are dramatically diminished [27,28]. Patients who have had AHSCT are more susceptible to virus reactivation and develop more severe viral infections [29]. International and national associations have published several guidelines for patients following stem cell transplantation. Patients should be immunized against pneumococci, Haemophilus influenzae, meningococcus, and influenza [23,24,25].

Our results indicate that a low hemoglobin level was related to a higher risk of infection development. Anemia in MM not only reflects disease severity but also affects quality of life, performance status, and cardiovascular health, possibly leading indirectly to a weakened immune system [30]. Accordingly, anemia was reported as a factor predisposing to various types of infections in different MM patient groups. In a cohort of newly diagnosed MM patients, Lin et al. found anemia (Hgb < 90 g/L) among advanced stage (ISS III) and elevated CRP to be risk factors associated with infection [31]. The occurrence of infection also contributed to inferior survival, such as factors influencing infection occurrence. Dumontet et al. included hemoglobin together with Eastern Cooperative Oncology Group (ECOG) performance status, lactate dehydrogenase, and serum β2-microglobulin level into the prediction model of the first treatment-emergent (TE) grade 3 infection in the first four months of treatment in MM patients [32].

A multivariate analysis of a smaller cohort of MM patients treated with the Rd regimen found lower Hb (<100 g/L) to be an independent factor associated with the occurrence of infections, together with the number of circulating CD3+CD4+CD161+ cells prior to Len-dex treatment [33]. The authors discovered that the existence of preexisting CD3+CD4+CD161+ cells might play a significant role in reducing the risk of severe infection in patients with RRMM after Rd treatment. However, the decline in hemoglobin content decreases the concentration of respiratory enzymes and mitochondrial oxidase, leading to an oxygen deficiency and the formation of lactic acid. Furthermore, these changes influence the immunological response and phagocytosis, which results in the suppression of immune activities and disruptions in immune regulation, raising the risk of infection [34]. Lactate has an immunosuppressive effect on T-cell proliferation, cytokine secretion, and the cytotoxic activities of NK and CD8+ T cells [35]. Similar to other studies, our study found a significant association between low hemoglobin levels and infection.

Recently, more direct mechanisms have been found between anemia and immunity impairment. Zhao et al. found anemia accompanied by a substantial deficit of CD8+ T cell responses against pathogens in untreated mice with large tumors [36]. This study identified a significant population of immunosuppressive cells, i.e., CD45+ erythroid progenitor cells (CD71+TER119+, EPCs); this may well contribute to the decreased T cell responses often reported in late-stage cancer patients. CD45+EPCs, following activation by tumor growth-associated extramedullary hematopoiesis, aggregate in the spleen to become the predominant population, outnumbering regulatory T cells (Tregs) and myeloid-derived suppressor cells (MDSCs). Like MDSCs, CD45+ EPC-mediated immunosuppression is mainly driven by reactive oxygen species generation. An immunosuppressive CD45+ EPC population was also found in anemic cancer patients [36].

In our analyses, hypoalbuminemia was found to be an independent factor influencing infection occurrence. Hypoalbuminemia is related to the development and severity of infectious illnesses, and robust innate and adaptive immune responses rely on albumin [37]. Serum albumin level in MM patients is an important prognostic factor, and together with β2-microglobulin, is included in the ISS score [38]. Patients with hypoalbuminemia present with lower hemoglobin level, poorer performance status, and larger disease burden assessed by bone marrow plasma cell infiltration [39]. Recently, using a pooled analysis of four clinical trials involving 1347 patients, the Spanish Myeloma Group proposed a simple risk score to predict early severe infection [40]. The score consisted of four variables: hypoalbuminemia (≤30 g/L), ECOG > 1, male sex, and non-IgA type MM. Tsai et al. reported in a large cohort of MM patients that light chain disease (HR 6.74), severe anemia (Hb < 80 g/L) (HR 3.34), serum hypoalbuminemia (HR 3.24), and allogenic SCT (HR 5.98) were independent predictors of invasive fungal infections in MM patients [41].

Recently, IMWG established recommendations for infection prevention in MM patients [42]. Patients with intermediate or high risk should receive levofloxacin for bacterial prophylaxis. In case of fungal infections, fluconazole or micafungin are indicated in cases of severe mucositis and prolonged neutropenia. Oral acyclovir for herpes simplex virus and herpes zoster virus is also recommended. Especially, patients receiving IMiD-based therapies should receive levofloxacin and acyclovir. Appropriate dose adjustment to prevent cytopenia occurrence in MM patients with renal involvement is required [42]. Patients with severe hypogammaglobulinemia or hypogammaglobulinemia with a history of life-threatening or recurrent infection are typical candidates for immunoglobulin replacement therapy. However, there are few trials examining the use of immunoglobulin replacement in patients with multiple myeloma, and most of these studies were conducted before the emergence of IMiDs [42]. MM patients with identified risk factors during IMiD-based treatment may potentially benefit from IVIG treatment.

In our cohort, we observed eight cases of confirmed COVID-19 infection and two of them required hospitalization. In a recent study, Martinez-Lopez et al., using global data from health care organizations, found that MM patients had a greater probability of SARS-CoV-2 infection and a higher excess death rate in 2020 than patients without MM. This study emphasizes the importance of expanding preventative measures globally to protect vulnerable individuals from SARS-CoV-2 infection by increasing social distancing and intensive immunization programs [43]. According to European Myeloma Network recommendations, the COVID-19 vaccine should be administered to all patients with monoclonal gammopathy of unknown importance, smoldering MM, MM, and monoclonal gammopathies of clinical relevance, as well as their family members. Patients should be vaccinated whenever feasible during stages of well-controlled illness and without concurrent anti-myeloma medication [44]. In patients with poorly controlled illness or ongoing therapy, vaccination may be undertaken on an individual basis, although the development of a protective immune response is less likely.

In our present study, the cohort median PFS was 12.6 months, slightly longer than reported in the pivotal Rd studies MM-009 and MM-010, in which pooled analysis reported a median PFS of 11.1 months [19]. In these studies, the majority (81.6%) of patients received at least two previous regimens, which undoubtedly influenced the outcome. Our results are also comparable to those from the standard arm of recent randomized controlled trials (RCTs) that used Rd doublet to test the addition of novel drugs, including daratumumab (POLLUX), carfilzomib (ASPIRE), or ixazomib (TOURMALINE); these reported a PFS of 14.7–17.6 months [45,46,47]. Real-world results are typically inferior to those of RCTs, since they frequently employ restrictive eligibility criteria excluding numerous subpopulations, resulting in poorer results of external validation [48]. Taken together, our outcome results seem to be reliable. One large (N = 290) real-life Italian study on Rd treatment in RR MM reported a time to progression of 11 months and a similar ORR [49]. Regarding infection occurrence frequency, direct comparisons are challenging. Firstly, due to our methodology based on medical chart review, we identified infections only when oral or intravenous therapy (antibiotic, antifungal, or antiviral intervention), hospitalization, and/or interruption of Rd medication were necessary, corresponding to grade ≥2 infections. Consequently, we noted a significantly lower rate of upper respiratory infections than RCTs [45,46,47]. However, the rate of pneumonia was generally similar to that reported in MM-009, MM-010, or POLLUX trials [19,45].

Our study has several limitations. Firstly, it only includes a retrospective analysis based on a revision of medical charts, and no detailed data could be obtained regarding severity, microbiological tests, or treatment. Secondly, in our cohort, several investigations, including CMV virus reactivation or immunoparesis, were not included in our analyses due to missing data regarding CMV-IgM and CMV-IgG and/or CMV antigenemia and levels of uninvolved immunoglobulins. In addition, cytogenetics data were only available for only a small fraction of our cohort, which could diminish its importance in the survival analysis. Nevertheless, we provide a cohort with an outcome similar to those of comparable studies.

## 5. Conclusions

AHSCT before Rd regimen therapy, hypoalbuminemia, and anemia during treatment were identified as three independent factors influencing the frequency of infections during Rd therapy in a large cohort of RRMM patients. Patients with known risk factors may benefit from optimal supportive therapy, which includes erythropoietin-stimulating medications and antimicrobial prophylaxis.

## Figures and Tables

**Figure 1 jcm-11-05908-f001:**
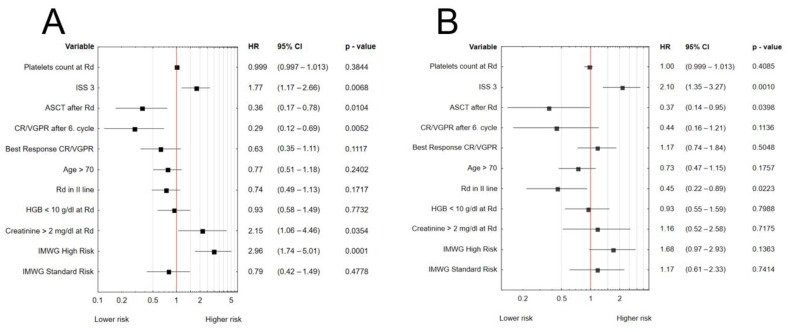
Multivariate Cox regression analyses for PFS (**A**) and OS (**B**).

**Figure 2 jcm-11-05908-f002:**
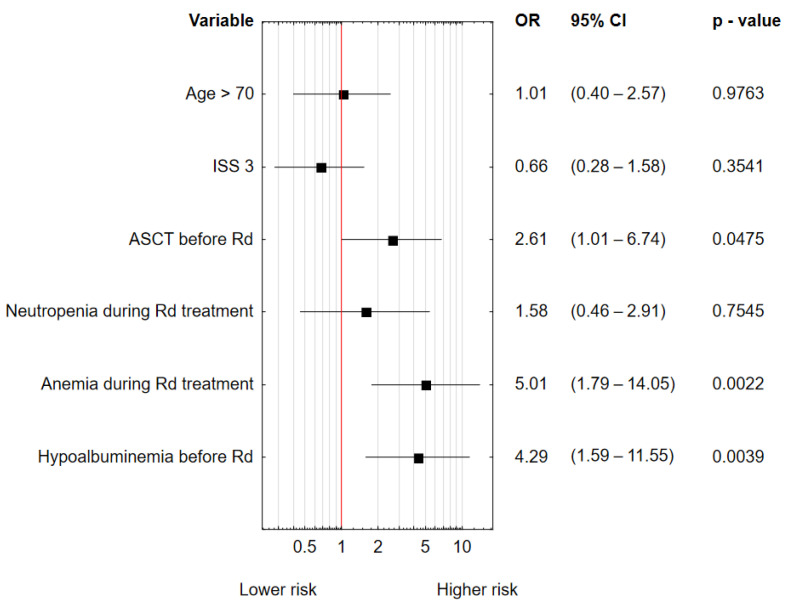
Multivariate logistic regression analysis for infection occurrence during Rd treatment.

**Table 1 jcm-11-05908-t001:** The characteristics of the MM patients included in the analysis. Data are shown as frequencies N (%) unless otherwise specified.

Characteristics	Total
Number of patients	174 (100%)
Sex	
M	84 (48.3)
F	90 (51.7)
Age at diagnosis	
Median (IQR)	65 (59.0–71.0)
Age at Rd administration	
Median (IQR)	68 (62.0–74.0)
Duration of Rd treatment (months)	
Median (IQR)	8.8 (4.3–18.24)
Myeloma stage:	
ISS I	37 (21.3)
ISS II	44 (25.3)
ISS III	72 (40.8)
Missing data	21 (12.1)
Transplant eligibility:	
AHSCT before Rd	57 (32.8)
AHSCT after Rd	16 (9.2)
Without AHSCT	109 (62.6)
Double AHSCT	16 (9.2)
Lenalidomide administration:	
Second-line	110 (63.2)
Third-line	57 (32.8)
Fourth-line	7 (4.0)
Adverse events:	
Infections	54 (31.0)
Neutropenia grade III and IV	50 (28.7)
Anemia grade III and IV	41 (23.6)
Thrombocytopenia grade III and IV	32 (18.4)
Pancytopenia	20 (11.5)
Thrombosis	20 (11.5)
Polyneuropathy	15 (8.6)
Nephrotoxicity	9 (5.2)
Cause of early ending of treatment	
Referral to AHSCT	16 (9.2)
Patient’s resignation	10 (5.7)
Hematological toxicity	8 (4.6)
Other nonhematological toxicity	4 (2.3)
Paraprotein type	
IgG	115 (66.1)
IgA	30 (17.2)
LCD kappa	14 (8.0)
LCD lambda	7 (4.0)
Biclonal	3 (1.7)
Nonsecretory	2 (1.1)
First-line treatment	
VCD	92 (52.9)
MP/MPT	16 (9.2)
CTD	30 (17.2)
VTD	8 (4.6)
Other	28 (16.1)
CRAB symptoms at Rd administration	
Ca > 2.5 mmol/L	24 (13.8)
Creatinine > 2 mg/dL	14 (8.0)
HGB < 100 g/L	39 (22.4)
Bone disease	95 (54.6)
RTx	85 (48.9%)
Cytogenetics *	52 (100%)
1q gain	21 (40.4)
Trisomies	11 (21.2)
del(13q)	10 (19.2)
t(4;14)	8 (15.4)
del(17p)	7 (13.5)
t(11;14)	2 (3.8)
t(14;16)	1 (1.9)

***** Cytogenetics data were available for 52 (29.9%) patients. Abbreviations: AHSCT—autologous hematopoietic stem cell transplant; Ca—Calcium; CTD—cyclophosphamide, thalidomide, dexamethasone; HGB—hemoglobin; IQR—interquartile range; LCD—light chain disease; MP/MPT—melphalan, prednisone/melphalan, prednisone, thalidomide; RTx—radiotherapy; VCD—bortezomib, cyclophosphamide, dexamethasone; VTD—bortezomib, thalidomide, dexamethasone.

**Table 2 jcm-11-05908-t002:** Univariate Cox regression for PFS and OS.

Parameter	PFS					OS				
	Coefficient	*p*-Value	HR	95% CI		Coefficient	*p*-Value	HR	95% CI	
				Lower	Upper				Lower	Upper
Sex (M)	0.030	0.7453	1.061	0.742	1.517	0.113	0.2597	1.255	0.846	1.862
Age > 70	0.094	0.3139	1.207	0.837	1.740	0.216	0.0357	1.541	1.029	2.306
ISS 3	0.276	0.0052	1.737	1.179	2.560	0.295	0.0066	1.804	1.179	2.761
AHSCT before Rd	−0.133	0.1828	0.767	0.519	1.133	−0.195	0.0842	0.677	0.435	1.054
AHSCT after Rd	−0.406	0.0270	0.444	0.216	0.912	−0.617	0.0159	0.291	0.107	0.794
Rd in II line	−0.162	0.0787	0.723	0.504	1.038	−0.147	0.1499	0.746	0.500	1.112
CR/VGPR after three cycles	−0.279	0.1514	0.572	0.266	1.227	−0.503	0.0487	0.365	0.134	0.994
CR/VGPR after six cycles	−0.503	0.0010	0.366	0.200	0.667	−0.451	0.0101	0.405	0.204	0.806
Best Response CR/VGPR	−0.451	0.0001	0.406	0.262	0.630	−0.576	0.0000	0.316	0.185	0.542
Ca > 2.5 mmol/L at Rd	−0.001	0.9918	0.997	0.586	1.697	0.071	0.6357	1.153	0.639	2.082
HGB < 100 g/L at Rd	0.188	0.0856	1.458	0.949	2.240	0.123	0.3069	1.278	0.798	2.045
Creatinine > 2 mg/dL at Rd	0.298	0.0605	1.816	0.974	3.387	0.240	0.1511	1.616	0.839	3.113
Bone disease at Rd	−0.161	0.1200	0.724	0.482	1.088	−0.060	0.6053	0.887	0.564	1.396
M protein concentration (g/L) at Rd	0.004	0.6602	1.004	0.987	1.021	0.008	0.3830	1.008	0.990	1.026
B2M concentration (mg/L) at Rd	0.022	0.3461	1.022	0.977	1.069	0.041	0.0534	1.042	0.999	1.086
LDH concentration (U/L) at Rd	0.001	0.1585	1.001	0.999	1.003	0.001	0.2613	1.001	0.999	1.004
WBC (×10^3^/μL) at Rd	0.029	0.3246	1.029	0.972	1.090	0.026	0.3639	1.026	0.971	1.085
PLT (×10^3^/μL) at Rd	−0.003	0.0194	0.997	0.995	1.000	−0.003	0.0359	0.997	0.995	1.000
Cytogenetic risk group										
Unknown	Reference					Reference				
High-risk	0.520	0.0019	1.954	1.242	3.075	0.300	0.0987	1.498	0.908	2.471
Standard-risk	−0.370	0.0574	0.803	0.456	1.414	−0.196	0.3381	0.913	0.505	1.650

Abbreviations: AHSCT—Autologous hematopoietic stem cell transplantation; B2M-Beta-2 microglobulin; Ca—Calcium; CR—Complete response; HGB—hemoglobin; ISS—International Staging System; LDH—lactate dehydrogenase; PLT—platelet count; RD—lenalidomide and dexamethasone; VGPR—very good partial response; WBC—white blood cell (WBC) count.

**Table 3 jcm-11-05908-t003:** Univariate logistic regression for infection during Rd treatment.

Parameter	Coefficient	*p*-Value	HR	95% CI	
				Lower	Upper
Age > 70	−0.184	0.5888	0.832	0.426	1.623
ISS 3	−0.156	0.6572	0.855	0.429	1.706
AHSCT before Rd	0.618	0.0708	1.855	0.949	3.625
AHSCT after Rd	−0.529	0.4320	0.589	0.157	2.205
Rd in II line	−0.195	0.5624	0.823	0.426	1.591
CR/VGPR after three cycles	0.221	0.7051	1.247	0.397	3.915
CR/VGPR after six cycles	0.113	0.8093	1.120	0.447	2.801
Best Response CR/VGPR	0.615	0.0870	1.850	0.915	3.741
Ca > 2.5 mmol/L at Rd	−0.194	0.6893	0.824	0.318	2.133
HGB < 100 g/L at Rd	0.049	0.9022	1.050	0.482	2.287
Creatinine > 2 mg/dL at 2 Rd	−0.145	0.8144	0.865	0.259	2.896
Bone disease at Rd	−0.240	0.5224	0.787	0.377	1.641
M protein concentration (g/L) at Rd	0.003	0.8427	1.003	0.974	1.033
B2M concentration (mg/L) at Rd	−0.009	0.8178	0.991	0.918	1.070
LDH concentration (U/L) at Rd	0.001	0.6299	1.001	0.996	1.006
Hypoalbuminemia	0.916	0.0257	2.500	1.118	5.593
Neutropenia grade 2 at Rd initiation	0.186	0.7287	1.205	0.421	3.448
Lymphocytopenia ≥ grade 2 at Rd initiation	−0.744	0.2012	0.475	0.152	1.487
WBC (×10^3^/μL) at Rd	−0.053	0.3148	0.949	0.856	1.051
PLT (×10^3^/μL) at Rd	0.002	0.3253	1.002	0.998	1.006
HGB (g/dL) at Rd	0.109	0.1919	1.115	0.947	1.312
Pancytopenia during Rd treatment	0.194	0.6979	1.214	0.455	3.239
Anemia grade ≥3 during Rd treatment	1.286	0.0006	3.618	1.741	7.521
Neutropenia grade ≥3 during Rd treatment	0.804	0.0221	2.234	1.122	4.448
Thrombocytopenia grade ≥3 during Rd treatment	0.177	0.6693	1.194	0.530	2.691

Abbreviations: AHSCT—Autologous hematopoietic stem cell transplantation; B2M-Beta-2 microglobulin; Ca—Calcium; CR—Complete response; HGB—hemoglobin; ISS—International Staging System; LDH—lactate dehydrogenase; PLT—platelet count; RD—lenalidomide and dexamethasone; VGPR—very good partial response; WBC—white blood cell (WBC) count.

## Data Availability

The data presented in this study are available from the corresponding author for request.

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
