# Peer review of "Risk Factors of Infection in Relapsed/Refractory Multiple Myeloma Patients Treated with Lenalidomide and Dexamethasone (Rd) Regimen: Real-Life Results of a Large Single-Center Study"

_jcm, 2022, doi:10.3390/jcm11195908_

Round 1

Reviewer 1 Report

Mikulski and colleagues present here real life data on infection risk in patients with MM receiving Rd as second (or above) line treatment.

Data are overall well presented, however I have some comments

1-     Data on survival outcomes are relevant, however I think the body of this section can be compressed and related figures moved to a supplementary section.

2-     I would add some details on the type of infections found in your cohort: did you find any fungal infections, CMV reactivation, etc? In how many cases the episode required hospitalization? Were all patients receiving (and compliant with) anti-infectious prophylaxis?

3-     How many Rd cycles were given prior to infection episode? How many patients had prior infection history, and did this increase the risk of infection during Rd?

4-     Do patients after HSCT routinely received vaccines (other than Covid vaccine)? Are immune reconstitution data available?

5-     I would also compress the discussion section from 252 to 274 and describe more how do you think the results of this study should impact clinical care.

Author Response

Reviewer 1.

Mikulski and colleagues present here real life data on infection risk in patients with MM receiving Rd as second (or above) line treatment.

Data are overall well presented, however I have some comments

  • Data on survival outcomes are relevant, however I think the body of this section can be compressed and related figures moved to a supplementary section.

    Response: The table with Univariate Cox regression analyses was reduced and KM plots were transferred to Supplementary Materials, according to the Reviewer’s suggestion.

2-     I would add some details on the type of infections found in your cohort: did you find any fungal infections, CMV reactivation, etc? In how many cases the episode required hospitalization? Were all patients receiving (and compliant with) anti-infectious prophylaxis?

Response: In our cohort, we did not observe any fungal infections. Regarding CMV virus reactivation, in our study, CMV-IgM and CMV-IgG and/or antigenemia tests were ordered in only 14 patients at various time points (before Rd, during Rd treatment, or after Rd before planned AHSCT). As these data were limited, they were not included in the manuscript.

We noted that 24 of 54 observed infections required hospitalization. Most patients received anti-infectious prophylaxis, including acyclovir (84.1%), 42.0% received sulfamethoxazole/trimethoprim, and 18.2% received fluoroquinolones including ciprofloxacin or levofloxacin.

This point was added to the manuscript.

3-     How many Rd cycles were given prior to infection episode? How many patients had prior infection history, and did this increase the risk of infection during Rd?

Response: The median number of cycles before the infection occurred was 2 (IQR 1-5 cycles). This information was added to the manuscript. Unfortunately, the past medical history addressing previous episodes of infections is difficult to acquire in a short period of time and was therefore omitted from the manuscript.

4-     Do patients after HSCT routinely received vaccines (other than Covid vaccine)? Are immune reconstitution data available?

Response: All patients who underwent AHSCT received vaccines according to the local recommendations [1] and international guidelines [2,3], including influenza, hepatitis B, pneumococci and Haemophilus influenzae. Regrettably, immune reconstruction data are not available. 

  1. Hus, A. Piekarska, J. RoliÅ„ski, K. Brzeźniakiewicz-Janus, K. Giannopoulos, K. Jamroziak, et al., Vaccination of adult patients with hematological malignancies and patients with asplenia – guidelines of PTHiT and Infectious Diseases Working Group PALG, Acta Haematol. Pol. 49 (2018) 93–101. https://doi.org/10.2478/ahp-2018-0016.
  2. Ljungman, C. Cordonnier, H. Einsele, J. Englund, C.M. Machado, J. Storek, et al., Vaccination of hematopoietic cell transplant recipients, Bone Marrow Transplant. 44 (2009) 521–526.
  3. Ludwig, M. Boccadoro, P. Moreau, J. San-Miguel, M. Cavo, C. Pawlyn, et al., Recommendations for vaccination in multiple myeloma: a consensus of the European Myeloma Network, Leukemia. 35 (2021) 31–44. https://doi.org/10.1038/s41375-020-01016-0.

5-     I would also compress the discussion section from 252 to 274 and describe more how do you think the results of this study should impact clinical care.

Response: We believe that our data revealing that anemia is an independent infection risk factor in RR MM patients treated with Rd are fairly interesting, and that the potential explanations for this in the above-mentioned discussion section enrich the manuscript (eg. Reviewer 2. opinion). According to the Reviewer's recommendation, potential implications have been included to the appropriate section.

Reviewer 2 Report

This is a very well presented Real-life study of risk factors of infection in relapsed/refractory multiple myeloma patients treated with lenalidomide and dexamethasone. Its most significant limitation is the retrospective nature of such studies, including incomplete datasets, but the authors recognise this and have produced a worthwhile manuscript which includes a particularly erudite Discussion.

Minor points include:

Line 4 of Introduction: Myeloma frequently develops from age 60 onwards rather than at age 60.

It is more common now to use SI units for Hb ie 90g/L rather than 9.0g/dL (though former still may be acceptable to Editorial Office).

The absence of data on acquired recurring cytogenetic aberrations in 70% of patients is a significant weakness. iFISH at presentation, and preferably also at relapse, should be a routine test in myeloma. In addition to contributing to risk stratification of patients, it may also provide worthwhile information on the underlying biological variations in the malignant plasma cells. Increasingly, this has the potential to inform treatment decisions, such as t(11;14) predicting response to venetoclax. 

There was an intriguing reference to CD161 in the Discussion, leaving the reader wishing for more information on, for example, serum IL-17 levels, T cell subset, B cell and NK cell levels in this patient cohort. This would have moved the study from one which was primarily observational, to a study which started to unravel immunologically significant factors underlying increased susceptibility to infections in relapsed/refactoriy myeloma patients treated with lenalidomide and dexamethasone.

However, despite the comments above, this is a very worthwhile study. The existence of an obviously strong clinical collaborative group should encourage the authors to continue with this work and supplement their studies with well designed, prospective data collection, when appropriate.

Author Response

Reviewer 2.

This is a very well presented Real-life study of risk factors of infection in relapsed/refractory multiple myeloma patients treated with lenalidomide and dexamethasone. Its most significant limitation is the retrospective nature of such studies, including incomplete datasets, but the authors recognise this and have produced a worthwhile manuscript which includes a particularly erudite Discussion.

Minor points include:

Line 4 of Introduction: Myeloma frequently develops from age 60 onwards rather than at age 60.

Response: The sentence was corrected accordingly.

It is more common now to use SI units for Hb ie 90g/L rather than 9.0g/dL (though former still may be acceptable to Editorial Office).

Response: Corrected as requested.

The absence of data on acquired recurring cytogenetic aberrations in 70% of patients is a significant weakness. iFISH at presentation, and preferably also at relapse, should be a routine test in myeloma. In addition to contributing to risk stratification of patients, it may also provide worthwhile information on the underlying biological variations in the malignant plasma cells. Increasingly, this has the potential to inform treatment decisions, such as t(11;14) predicting response to venetoclax. 

Response: The authors fully agree with the Reviewer's arguments. In Poland, the availability of cytogenetic testing for multiple myeloma was limited in previous several years. Even if obtained, cytogenetics had minimal impact on therapeutic decisions in routine clinical practice due to the absence of novel drugs. Then, outside of clinical trials, only younger "fit" patients underwent cytogenetic analysis as part of the qualification process for standard/tandem AHSCT. Recent reimbursement of novel drugs in Poland, including daratumumab (2019), carfilzomib (2019), and ixazomib (2021), has led to routine cytogenetics testing.

There was an intriguing reference to CD161 in the Discussion, leaving the reader wishing for more information on, for example, serum IL-17 levels, T cell subset, B cell and NK cell levels in this patient cohort. This would have moved the study from one which was primarily observational, to a study which started to unravel immunologically significant factors underlying increased susceptibility to infections in relapsed/refactoriy myeloma patients treated with lenalidomide and dexamethasone.

Response: Although a detailed analysis of how various immunological factors contribute to the occurrence of infections in relapsed/refractory MM patients would be a fascinating study, it would likely require a different experimental approach that is far beyond the scope of this retrospective study. Nevertheless, we have made some preparations to conduct a prospective study of several potential biomarkers that enable the identification of patients at high risk for infection in the near future.

However, despite the comments above, this is a very worthwhile study. The existence of an obviously strong clinical collaborative group should encourage the authors to continue with this work and supplement their studies with well designed, prospective data collection, when appropriate.

Reviewer 3 Report

In this study the investigators examine risk factors of infections in RRMM patients treated with Rd in a real-world patient cohort from a large single center. The study is interesting and well designed with appropriate statistics, and the reporting of real-world data is important. However, there are a few comments which should be addressed before acceptance of the paper.

Introduction

1.    Line 42 to 43: The median age at diagnosis in population-based studies/ registries is around 70 years (PMID: 33498356, https://www.cancerresearchuk.org/health-professional/cancer-statistics/statistics-by-cancer-type/myeloma/incidence#heading-One)

2.       Lines 48 to 60 seem irrelevant to the topic – suggest adding how this is connected to the topic or remove.

3.       Add clinical trials that led to the approval of Rd and the risk of infections reported in the trials.

Results

1.       You report that ASCT after Rd is an independent protective prognostic factor for PFS and OS but this could be explained by immortal time bias. Did you take this into account? If not, this should be added as a possible bias.

2.       Line 185-186: Please add what variables were included in the multivariate logistic regression analysis.

3.       Line 187-189: You report that anaemia is an independent factor related to the occurrence of infections, was this independent of disease progression?

4.       Table 3: Did you not include immunoparesis in your analysis? If not, why? If not possible to add, this should be added as a limitation.

5.       Did any of the patients die from the infections (case fatality rate)?

Discussion

1.       Please relate the rate of infections in your study to the reported adverse events/infections in clinical trials using Rd, if not similar, why?

2.       You conclude that patients with known risk factors may benefit from EPO-stimulating medication and antimicrobial prophylaxis. Perhaps add something in discussion about a possible role of IV immunoglobulins.

Author Response

Reviewer 3. 

In this study the investigators examine risk factors of infections in RRMM patients treated with Rd in a real-world patient cohort from a large single center. The study is interesting and well designed with appropriate statistics, and the reporting of real-world data is important. However, there are a few comments which should be addressed before acceptance of the paper.

Introduction

  1. Line 42 to 43: The median age at diagnosis in population-based studies/ registries is around 70 years (PMID: 33498356, https://www.cancerresearchuk.org/health-professional/cancer-statistics/statistics-by-cancer-type/myeloma/incidence#heading-One)

Response: The sentence was corrected accordingly and the reference was added.

  1. Lines 48 to 60 seem irrelevant to the topic – suggest adding how this is connected to the topic or remove.

Response: This section was significantly reduced, according to the Reviewer’s suggestion.

  1. Add clinical trials that led to the approval of Rd and the risk of infections reported in the trials.

Response: Corrected as requested.

Results

  1. You report that ASCT after Rd is an independent protective prognostic factor for PFS and OS but this could be explained by immortal time bias. Did you take this into account? If not, this should be added as a possible bias.

Response: Immortal person-time bias occurs when the outcome of research participants is compared across groups defined by an event occurring during follow-up [1,2]. Using such definitions necessitates that persons who meet the criteria survive until the defining event takes place. From the view of the data analysis, the time between the start of follow-up and the defining event appears to be "immortal" for those who experience the event. When this 'immortal' person-time is factored into the denominator of the mortality rate, a large bias is created in favor of the group that experienced the event.

In our study, 16 patients (9.2%) underwent AHSCT after Rd treatment. The median time from Rd commencement and AHSCT in this group was 10.9 months. Despite this, median PFS in this group was over 35 months, and the median OS was not reached (corresponding KM plots are shown in Figure 2). Even if these findings are computed from the day of AHSCT, they are much better than the median PFS (12.6 months) and median OS (22.3 months) in our study. For this reasons we think that immortal bias is in our case is neglectable.

  1. Giobbie-Hurder A, Gelber RD, Regan MM. Challenges of guarantee-time bias. J Clin Oncol 2013; 31: 2963–2969.
  2. Hanley JA, Foster BJ. Avoiding blunders involving 'immortal time'. Int J Epidemiol 2014; 43: 949–961.
  3. Line 185-186: Please add what variables were included in the multivariate logistic regression analysis.

Response: As shown in Figure 3, age, ISS, ASCT before Rd, neutropenia and anemia during Rd treatment and hypoalbuminemia were included in multivariate analysis. Description of analysis was updated as requested.

  1. Line 187-189: You report that anaemia is an independent factor related to the occurrence of infections, was this independent of disease progression?

Response: In the analysis all cases of anemia grade 3 during Rd treatment were included. Occurrence of anemia was reported in 41 patients, of which 23 (56.1%) had further PD. On the other hand, the frequency of anemia in patients with disease progression and without disease progression was 27.1% and 20.5%, respectively (p=0.31).

  1. Table 3: Did you not include immunoparesis in your analysis? If not, why? If not possible to add, this should be added as a limitation.

Response: Due to retrospective nature of our study, there is serious missing data regarding levels of uninvolved immunoglobulins. This limitation was added in discussion accordingly.

  1. Did any of the patients die from the infections (case fatality rate)?

Response: In some cases, it was not possible to confirm the cause of death, such as when the patient was treated at another hospital. However, three patients (5.6%) died within 30 days of infection diagnosis, indicating a potential causative impact.

Discussion

  1. Please relate the rate of infections in your study to the reported adverse events/infections in clinical trials using Rd, if not similar, why?

Response: Regarding infection occurrence frequency, direct comparisons are challenging. Firstly, due to our methodology based on medical chart review, we identified infections only when oral or intravenous therapy (antibiotic, antifungal, or antiviral intervention), hospitalization, and/or interruption of Rd medication were necessary, corresponding to grade ≥2 infections. Consequently we noted a significantly lower rate of upper respiratory infections than RCTs [48-50]. However, the rate of pneumonia was generally similar to that reported in MM-009, MM-010, or POLLUX trials.

The appropriate section was added to the manuscript.

  1. You conclude that patients with known risk factors may benefit from EPO-stimulating medication and antimicrobial prophylaxis. Perhaps add something in discussion about a possible role of IV immunoglobulins.

Response: Patients with severe hypogammaglobulinemia or hypogammaglobulinemia with a history of life-threatening or recurrent infection are typical candidates for immunoglobulin replacement therapy. However, there are few trials examining the use of immunoglobulin replacement in patients with multiple myeloma, and most of these studies were conducted before the emergence of IMiDs [1]. MM patients with identified risk factors during IMiD-based treatment may potentially benefit from IVIG treatment.

This section was added to the manuscript. 

  1. S. Raje, E. Anaissie, S.K. Kumar, S. Lonial, T. Martin, M.A. Gertz, et al., Consensus guidelines and recommendations for infection prevention in multiple myeloma: a report from the International Myeloma Working Group, Lancet Haematol. 9 (2022) e143–e161. https://doi.org/10.1016/S2352-3026(21)00283-0.

Round 2

Reviewer 1 Report

The authors have responded to all questions, I have no further comments.